# G4mismatch: Deep neural networks to predict G-quadruplex propensity based on G4-seq data

**Mira Barshai**[1], **Barak Engel**[1�], **Idan Haim**[1�], **Yaron Orenstein**[1,2,3] *

**1** School of Electrical and Computer Engineering, Ben-Gurion University of the Negev, Beer-Sheva, Israel,
**2** Department of Computer Science, Bar-Ilan University, Ramat Gan, Israel, **3** The Mina & Everard Goodman
Faculty of Life Sciences, Bar-Ilan University, Ramat Gan, Israel

и These authors contributed equally to this work.
* yaron.orenstein@biu.ac.il

Perspectives Organization, SUDAN

**Data Availability Statement:** All G4-seq data used
for model fitting and testing was acquired from
GEO accession code GSE110582. Microarray data
used for model validation, was downloaded from
https://pubs.acs.org/doi/suppl/10.1021/

## Abstract

G-quadruplexes are non-B-DNA structures that form in the genome facilitated by Hoogsteen
bonds between guanines in single or multiple strands of DNA. The functions of G-quadru-
plexes are linked to various molecular and disease phenotypes, and thus researchers are
interested in measuring G-quadruplex formation genome-wide. Experimentally measuring
G-quadruplexes is a long and laborious process. Computational prediction of G-quadruplex
propensity from a given DNA sequence is thus a long-standing challenge. Unfortunately,
despite the availability of high-throughput datasets measuring G-quadruplex propensity in
the form of mismatch scores, extant methods to predict G-quadruplex formation either rely
on small datasets or are based on domain-knowledge rules. We developed G4mismatch, a
novel algorithm to accurately and efficiently predict G-quadruplex propensity for any geno-
mic sequence. G4mismatch is based on a convolutional neural network trained on almost
400 millions human genomic loci measured in a single G4-seq experiment. When tested on
sequences from a held-out chromosome, G4mismatch, the first method to predict mismatch
scores genome-wide, achieved a Pearson correlation of over 0.8. When benchmarked on
independent datasets derived from various animal species, G4mismatch trained on human
data predicted G-quadruplex propensity genome-wide with high accuracy (Pearson correla-
tions greater than 0.7). Moreover, when tested in detecting G-quadruplexes genome-wide
using the predicted mismatch scores, G4mismatch achieved superior performance com-
pared to extant methods. Last, we demonstrate the ability to deduce the mechanism behind
G-quadruplex formation by unique visualization of the principles learned by the model.

## Author summary

G-quadruplexes (G4s) are non-canonical secondary structures, which have been exten-
sively studied, and found to be associated with numerous diseases. The G4-seq experiment
provided valuable data, mapping G4s across the genomes of 12 different species, reporting
the potential of a DNA region to form a G4 by a mismatch score. Previous methods to

acschembio.9b00934/suppl_file/cb9b00934_si_003.xlsx. G4mismatch code is available at https://github.com/OrensteinLab/G4mismatch/.

**Funding:** This work was partially supported by the Israeli Council for Higher Education (CHE) via Data Science Research Center, Ben-Gurion University of the Negev, Israel https://che.org.il/ to YO, and by the Israel Science Foundation (grant no. 358/21) to YO. The funders had no role in study design, data collection and analysis, decision to publish, or preparation of the manuscript.

predict G4s simply solved the problem of G4-folding as binary classification or focused on putative quadruplexes rather then predicting the raw genome-wide scores generated by the G4-seq experiment.

Our new approach, G4mismatch, is the first to utilize millions of G4 mismatch scores measured by the G4-seq experiment as a highly accurate simulator of a G4-seq experiment, which can predict the mismatch score of any given DNA sequence and by that uncover its potential to form a G4. In addition, our work utilizes data from all 12 different species to demonstrate the ability of a model trained on one species to predict on other genomes, and explore the properties that give advantage to some models over others. Moreover, we show how the model learned known and novel molecular principles underlying G4 folding.

This is a *PLOS Computational Biology* Methods paper.

## Introduction

The formation of DNA secondary structures can influence biological processes such as replication, translation, and splicing [1, 2]. Single-stranded DNA undergoes base-pairing between stretches of guanine nucleotides (G-tracts) facilitaed by Hoogsteen bonds to form four-stranded structures known as G-quadruplexes (G4s) [3–5] that occur *in vivo* [6]. G4 secondary structures arise in G-rich sequences where four guanine bases interact to form planar G-tetrads (Fig 1A), which can self-stack [7] (Fig 1B). Nucleotide sequences containing four tracts of three or more guanines separated by loops of variable length can also form G4s spontaneously

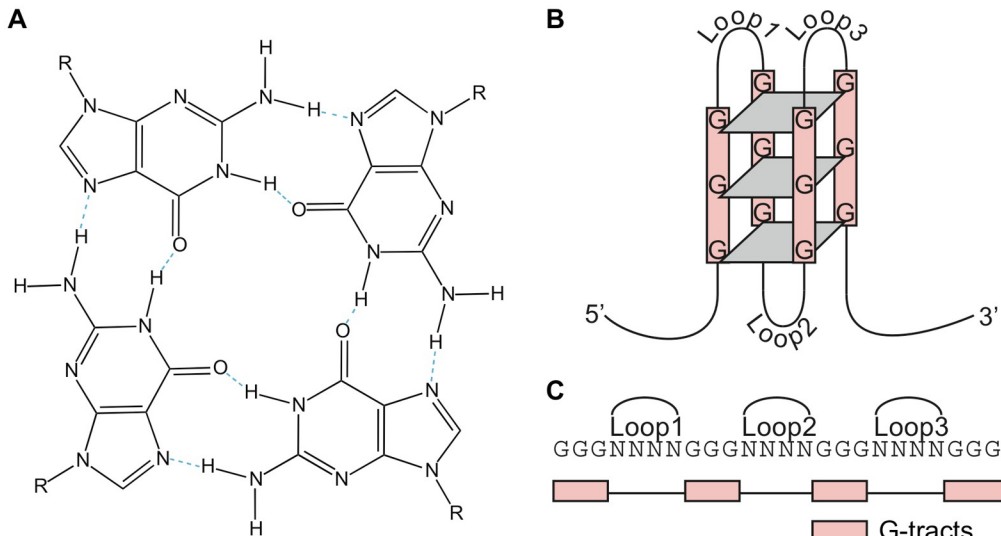

**Fig 1. G-quadruplex (G4) structure.** (A) Schema of a G-tetrad that makes up the core of a G4 structure, which are stabilized by coordination of an alkali cation (blue). (B) Schematic representation of a parallel unimolecular G4. (C) Example of a single-stranded DNA G4 motif, consisting of four G-tracts (pink), separated by three 4 nt-long loops. Figure adapted with permission from [12] under a CC BY license.

*in vitro* and are stabilized by an alkali cation, such as potassium or sodium [5, 8–10] (Fig 1C). The known types of DNA sequences that can form G4s are expanding, and several non-canonical G4s have been described including those with longer loops, bulges, or mismatches in their G-tracts [11]. G4 formation is kinetically fast, and these structures are thermodynamically stable under physiological conditions, particularly in the presence of $K^+$ [7].

Computational predictions using simple algorithms suggest that over 300,000 sequence motifs of the type

$$G^{3+}[ACGT]^{1-7}G^{3+}[ACGT]^{1-7}G^{3+}[ACGT]^{1-7}G^{3+} \tag{1}$$

in the human genome have the potential to form a G4 structure [13, 14], denoted as putative quadruplexes (PQs). A more recent algorithm predicted that the number of potential G4 sequences is substantially higher [15]. These computational studies predict that G4 motifs are enriched in telomeres, promoters, and the first intron of genes, but they highlight the need to generate explicit experimental data about the existence and functions of G4s in biologically relevant contexts.

Several experimental techniques were developed to measure G4 formation in high-throughput [16, 17]. G4 formation in a DNA template can be assessed using polymerase stop assays, which measure polymerase stalling at G4 sites [18]. A method, called G4-seq, was developed by combining features of the polymerase stop assay with Illumina next-generation sequencing [19]. In this technology, high-throughput DNA sequencing is performed under conditions that either promote or disfavor G4 formation. Because polymerase stalling at G4 sites was found to affect base calling, sequencing readouts from both conditions can be compared to elucidate the exact position of G4 structures. The experimental output is a partition of the genome to 15 nt-long bins with a mismatch score assigned to each bin. Two approaches were separately used to promote DNA G4 stabilization: adding $K^+$ or adding $K^+$ with G4-stabilizing ligand pyridostatin (PDS) to the sequencing buffers [20].

Various computational methods have been developed for the task of G4 detection [21]. Over the years, the canonical expression (Eq 1) served the prime mean for G4 detection, but experimental studies uncovered non-canonical G4-forming sequences [22–25]; thus, emphasizing the need for a more flexible G4 detection tool. Rule-based methods, such as G4Hunter [15, 26, 27], pqsfinder [28], and g4predict [29], enable the user to diverge from the strict rules imposed by the regular expression, but they are mostly based on domain knowledge rather than direct experimental data. G4Hunter applies a sliding window approach, but fails to consider the importance of the feature position within the sequence. pqsfinder penalizes potential G4s based on imperfections within the G-quartets and loop lengths, but ignores other associations between the nucleotides, which may interfere or encourage G4 formation. g4predict extended the canonical G4 definition by using user-defined parameters for loop and G-tract length. The detected sequences are scored based on the length of the tetrads and penalized based on loop length and the appearance of bulges. Quadron uses a tree-based gradient-boosting machine to assess the potential of a sequence to form a G4 using features from within the suspected sequence and from flanking sequences on both ends [30]. Quadron, though showing great predictive performance, was limited by training only to the canonical form of a G4. G4detector [31], PENGUINN [32], and DeepG4 [33] are sequence-based classification methods utilizing the latest advancement in machine learning, i.e. deep neural networks (DNNs) [34]. G4detector, PENGUINN, and DeepG4 use a convolutional neural network (CNN) to evaluate the probability of a given DNA sequence to form a G4, but unlike the first two methods, which were solely trained on *in vitro* data, DeepG4 was trained to identify G4s which are active *in vivo*. The use of a DNN reduces the need for feature engineering, which is one of the

key features of DNNs. Still, no method was developed as a genome-wide prediction model capable of predicting mismatches caused by polymerase stalling to any genomic sequence, which is a proxy to the stability of a G4.

Herein, we present *G4mismatch*, a CNN for predicting a quantitative value that correlates with G4 stability. We developed and trained G4mismatch to predict mismatch scores to any given DNA sequence, and with that we can predict the potential of any given sequence to form a G4. We compared the performance of G4mismatch to extant methods and the state of the art over the G4 detection task, and tested it on held-out datasets from human and other species. Last, we applied unique visualizations to highlight important features identified by G4mismatch in order to deduce insights underlying the G4-formation mechanism.

## Materials and methods

### Datasets

**G4-seq data.** In this study, we used the improved G4-seq dataset covering whole-genome G4 measurements over 12 species [20] (publicly available via accession code GSE110582), including widely studied model organisms and pathogens of clinical relevance. The experiment covers both forward and reverse DNA strands, and assigns to almost every non-overlapping 15 nt-long bin in each of the 12 genomes a mismatch score. The G4 mismatch score, as defined by the developers of the G4-seq assay, is calculated as the ratio of the number of mismatched base calls observed under a G4-stabilizing condition compared to control conditions over the length of the complete sequence (either mismatched or not). Hence, its range is 0% to 100%. G4 structures that form under physiological $K^+$ conditions were identified as well as G4s stabilized by a combination of $K^+$ and PDS. The numbers of sequence scores in the G4-seq dataset used in our study are summarized in S1 Table.

**G4-binding microarray data.** To validate G4mismatch on an independent dataset, we used a recently published microarray data measuring G4 binding of different molecules, among them PDS [35]. This study used three Agilent DNA microarray designs, which together contain a total of 24,154 unique sequences, to examine the binding specificity of proteins, antibodies, and small molecules to G4s and G4 variants. The binding strength was visualized by fluorescence imaging, validating the platform as a high-throughput method to profile G4-binding specificity. We used the PDS-binding dataset from that study under design 3, which includes 15,671 sequences and is the most comprehensive of the three designs, and includes G4s, their variants, and non-G4 sequences as controls. None of those sequences appeared in the G4-seq dataset.

### G4mismatch neural network architecture

G4mismatch to predict a mismatch score of a given DNA sequence is based on a popular CNN architecture in genomics [34] (Fig 2A). The input to G4mismatch is a one-hot-encoded 215 nt-long DNA sequence matrix. N positions in the DNA sequence were replaced by a vector of 0.25 indicating a uniform probability to all nucleotides. The CNN architecture includes a 1D-convolutional layer comprising 256 kernels of size 95, a global max-pooling layer, and a fully connected layer of 32 neurons with ReLU activation. The network's output goes through a single neuron with linear activation. The CNN outputs a predicted mismatch score for a given DNA sequence, thereby simulating a G4-seq experiment for that sequence. In the output neuron, we did not use a sigmoid function, which outputs a value between 0 to 1 which is the range of mismatch scores, for two reasons. First, since a mismatch score is only correlated to G4 stability (and not an absolute value of G4 stability), we are more interested in predicting a value that is highly correlated with a mismatch score rather than the exact value. Second, when evaluated on the validation set,

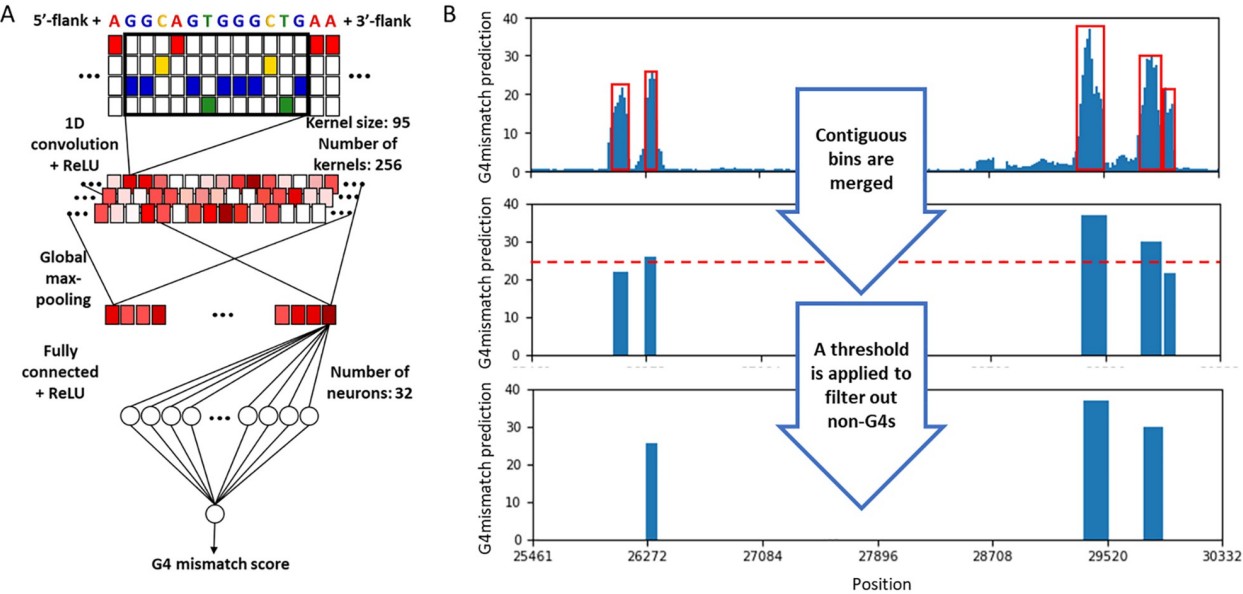

**Fig 2. G4mismatch for mismatch score prediction and G4 identification.** (A) For mismatch score prediction of a given 15 nt-long sequence, the sequence is padded with 100 nt-long flanks on each side and is one-hot encoded. The one-hot matrix goes through a 1D-convolutional layer with ReLU activation, consisting of 256 kernels of size 95. The output of the convolutional layer goes through a global max-pooling layer and a fully connected layer with 32 neurons. The network output is governed by a single neuron with linear activation. (B) The predicted mismatch scores are used to identify potential G4-forming genomic regions. Contiguous bins with a mismatch score above 14.4 are merged and assigned the maximum mismatch score within the merged area. Sequences with a score smaller than 25 and 35 for K$^+$ and K$^+$+PDS experiments, respectively, are filtered out. The resulting sequences are defined as having the potential to form a G4.

prediction performance decreased by 0.1 Pearson correlation when using a sigmoid activation function in the output neuron compared to using a linear activation function.

To evaluate the effect of flanking sequences and kernel size on the prediction performance of G4mismatch, we trained and tested the model with flanks of varying length from 0 to 140 nt with increments of 20 nt, and for each flanking sequence length $L$ all kernel sizes from 5 to $2L$ + 15 with increments of 10. To gauge the importance of the 5' compared to 3' flank of the 15 nt-long central bin, we trained a model with an input sequence with a flank on only one side. We then measured the prediction performance of each of the models (5' model compared to 3' model).

## Training G4mismatch models

Due to the large amount of data processed by G4mismatch during training (e.g., almost 400 million sequences and mismatch scores for the human genome), neither the one-hot-encoded sequences nor the sequences extracted from the genome were stored in memory during the training process. Rather, only the genomic coordinates, mismatch scores, and a one-hot encoding of the reference genome were stored in memory. During the training process, the sequences were extracted "on-demand" every time a new training batch was required. 13 CPUs simultaneously read and processed the data into mini-batches, feeding them to the GPU to avoid a bottleneck in the training process.

To train the model and select optimal hyper-parameters, chromosome 2 was excluded from the training dataset and used as a validation set. Chromosome 1 was held out to serve as a test set. The learning rate was set to 0.001 and batch size to 256. Due to the large size of the training set and its redundant nature, one training epoch was enough for proper model convergence.

We did not search for optimal hyper-parameters values, excluding the flanking sequence and kernel sizes, since the prediction performance was already excellent with our initial choice of hyper-parameters values. G4mismatch was implemented using Keras library with Tensorflow backend. We gauged prediction performance by Pearson correlation coefficient, which highlighted its ability to predict the relatively rare cases of high mismatch scores, as this metric is highly sensitive to outliers [36]. The model was trained using NVIDIA TITAN V GPU and Intel Xeon Silver 4110 CPU. Training a G4mismatch model (Fig 2A) on the $K^+$ human dataset with 100 nt-long flanking sequence on each side, required 8,588 seconds per epoch, 1338% CPU usage, and maximum memory usage of 82.73GB.

## Detection of G-quadrulex forming sequences

In addition to evaluating G4mismatch in predicting mismatch scores, we also aimed to demonstrate its performance in the G4 detection task. Following the mismatch score prediction, we applied the process performed in the G4-seq study to identify G4-forming sub-sequences [20]. This process merges contiguous genomic regions with a mismatch score greater than 14.4. This threshold was derived by us so that the process of producing peaks from the published list of 15 nt-long bins and corresponding mismatch scores resulted in the list of published peaks. Then, we assigned the maximum mismatch score in that region to the entire sequence. We used bedtools merge command to perform both the merge and score assignment [37]. The thresholds of 25 and 35 for $K^+$ and $K^+$+PDS were used to filter out non-G4s, respectively (Fig 2B).

## Running extant methods for G4 prediction

We tested the ability of G4mismatch to detect G4-forming sequences against trained models of various extant methods developed for same task:

- G4Hunter [15, 26, 27]

- pqsfinder [28]

- g4predict [29]

- Quadron [30]

- PENGUINN [32]

- DeepG4 [33]

We used G4Hunter with a window size of 25 and a threshold of 1, as suggested in [15]. We ran pqsfinder and g4predict with their default parameters. The threshold for pqsfinder was set to 47 as it was found to be the optimal threshold for G4 identification [38]. For Quadron a threshold of 25 was set to match the $K^+$ dataset and 35 for $K^+$+PDS dataset (we did not use the originally suggested threshold of 19, since it resulted in poorer performance). We used PENGUINN as a genome-wide scanner predicting a score for every 200 nt-long sequence in chromosome 1, similarly to the scan we performed with G4mismatch. We used two different thresholds for PENGUINN (as defined in the original study [32]) 0.5 and 0.85 denoted by PENGUINN(s) and PENGUINN(p), respectively, to filter the scan results. All methods were fed with chromosome 1 from the hg19 reference genome to identify the relevant sub-sequences on both forward and revers strands. Last, we scanned chromosome 1 with DeepG4 using DeepG4Scan with stride of size 1. In this case, we split choromosome 1 into 1000 nt-long sequences with an overlap of 201 nt, the input size of DeepG4. As no previous study offered a computational mismatch prediction method, we did not test against other methods in the mismatch score prediction task.

## Cross-species training and prediction analysis

The G4-seq dataset provided an unprecedented opportunity to test how G4-folding principles learned from data of one species transfer to other species. To test this, we performed a cross-species analysis. We trained a species-specific model for each species on its complete dataset. Then, we predicted mismatch scores on the other 11 species. For comparison purposes, all 12 models were trained in a similar manner, regardless of the properties of the training set. We calculated the Pearson correlation of measured and predicted mismatch scores for each species separately.

Given the cross-species results, we investigated why some experimental datasets led to more accurate models than others. For this aim, we measured the Spearman correlation of the G4mismatch correlation results to various properties of each training genome and experiment, which were available from the G4-seq study [20]. The properties we tested against were:

- Genome size

- GC-content (%GC)

- Number of extended PQ motifs (#PQs, loop length varying between 1 and 12)

- PQs density

- Single-nucleotide coverage

- Frequency of mismatch scores under 5% (mismatch<5%)

For each property, we combined the p-values of the Spearman correlations across the 11 other species by the harmonic mean p-value [39].

## Visualization of G4-folding principles

To deduce insights behind the molecular mechanism of G4 formation, we interrogated the G4mismatch-trained model. We first visualized the impact of loop length on predicted mismatch scores. We predicted mismatch scores for a canonical G4 with a variable loop lengths, while varying the length of each loop separately between 1 to 12. The loops and flanks were encoded such that each nucleotide at those positions was assigned an equal probability, i.e. 0.25.

Next, we explored the impact of mismatches in the G-tracts on the predicted mismatch score. For this aim, we defined the wild-type G4 to be: GGGNGGGNGGGNGGG. We predicted a mismatch score for each G4 variant with a mutation in one of the guanines in the G-tracts to all possible nucleotides. The impact of the mutation was measured as the difference between the predicted score of the wild-type G4 and the mutated G4.

Last, we evaluated how the nucleotide composition of the sequences flanking a 15 nt-long PQ affect mismatch-score prediction. For this aim, we generated synthetic sequences such that the probability of each nucleotide to appear in a 20 nt-long bin in a given flank varied from 0 to 1, while the other three nucleotides were assigned an equal probability. We applied this process to all non-overlapping 20 nt-long bins from positions adjacent to the central 15 nt-long bin of the form GGGNGGGNGGGNGGG to the edge of the flank.

Moreover, we used the Integrated Gradients (IG) approach to identify key features within a given sequence [40]. This approach attributes a DNN's prediction to features of the input relative to a neutral baseline and assigns a score to each such feature, which indicates its importance. The method computes the path integral of the gradients along a straight path between the input and the baseline. We used an all-0.25 matrix baseline, representing equal probability for each nucleotide in each position of the sequence. To visualize the preferences learned by the G4mismatch model by IG, we identified the maximum mismatch score over all 15 nt-long

sub-sequences of the sequence AGGGCGGTGTGGGAAGAGGGAAGAGGGGGGAGGCAG, extracted from the promoter region of the KRAS proto-oncogene, along with 100 nt-long flanks on each side, and visualised the attribution scores of that sub-sequence and its flanks. We calculated attribution scores using the implementation suggested in [41]. The size of the letters in our visualizations corresponds to the relative importance of the given position of the input, with positions more relevant to the final predictions having a greater size. We also used mutation maps to explore G4mismatch sensitivity to mutations across all positions. To generate the mutation map, we predicted the mismatch score of a wild-type sequence, and then systematically varied at each position to the other three nucleotides and measured the sensitivity by subtracting the predicted mismatch score of the mutated sequence from the score of the wild-type sequence.

## Results

### Whole-genome mismatch-score prediction evaluation

We developed G4mismatch, a novel method based on CNNs to predict a mismatch score for any given sequence. We trained G4mismatch to predict a mismatch score for any 15 nt-long DNA sequence appended by its upstream and downstream flanking sequences, termed flanks. We report the results in a leave-chromosome-out setting, which is a common approach previously performed for various genomic tasks, such as transcription-factor-binding prediction [42, 43], pre-miRNAs prediction [44], and 3'UTR-element prediction [45]. We trained on data from all chromosomes but the test and validation chromosomes, which were chromosomes 1 and 2, respectively. We evaluated prediction performance by Pearson correlation of predicted and measured mismatch scores.

**Flanking-sequence-length and kernel-size effect on mismatch-score prediction performance.** We concatenated flanking sequences to the 15 nt-long genomic sequences, corresponding to each bin for which a mismatch score was reported, as input to our model. The critical parameter for prediction performance, that we evaluated, was the length of the upstream and downstream flanking sequences. Longer flanks may include more information (either biological or experimental) but up to a certain point as distant sequences are less likely to form a G4 or affect G4 formation, and thus have a smaller effect on mismatches induced by G4 formation. Moreover, longer flanks increase model parameters and by that increase the risk of model overfitting.

We evaluated the effect of several different flank lengths starting from no flanks and increasing by 20 nt on each side until no further improvement in prediction performance was observed on the validation set (chromosome 2) (Fig 3A). For the $K^+$ stabilizer, prediction performance as measured by Pearson correlation increased from 0.38 with no flanks to 0.82 with 100 nt-long flanks. On the $K^+$+PDS dataset results were even better: increasing from 0.45 with no flanks to 0.91 with 100 nt-long flanks (S1(A) Fig). These results demonstrate the importance of the flanking information for G4 propensity prediction, as was observed previously by the developers of Quadron [30], and systematically tested by us for the mismatch genome-wide prediction task. It also points out that 100 nt is likely around the optimal flank length. We note that there is very little train-test data leakage. While some sequences may share similarities between chromosomes (and any two G4 sequences are expected to share similarity as they are based on multiple G-tracts), there are very few identical 215 nt-long sequences shared between chromosome 1 and the chromosomes comprising the training set (only 0.3% as we calculated by Jellyfish k-mer counter [46]).

In parallel to the evaluation of the optimal flanking sequence length, we evaluated the effect of the 1D-convolutional kernel size. For each flank length $L$, we varied the kernel size from 5

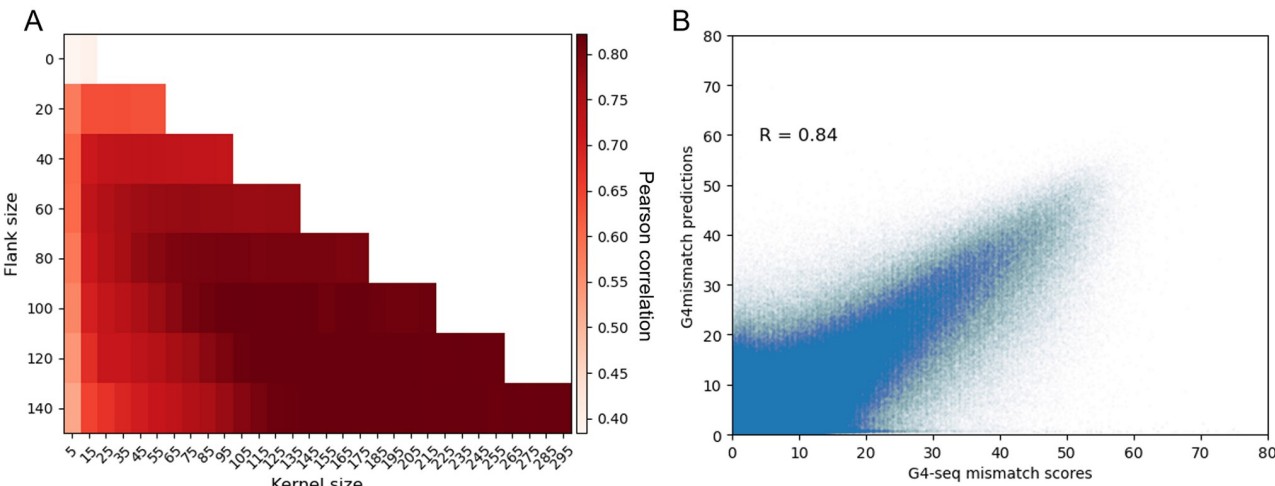

**Fig 3. G4mismatch K$^+$ model prediction performance.** (A) For each flanking sequence length, multiple models were trained differing in their kernel size. Increasing the kernel size resulted in improved performance, which plateaued at a kernel size of the flank length. (B) G4mismatch predictions on chromosome 1 test set highly correlate with G4-seq experimental measurements.

to $2L + 15$. Previous studies applying CNNs to genomic tasks used smaller kernel sizes (5–10) for prediction tasks. Using smaller kernels, while resulting in shorter training times, may be disadvantageous by missing relationships between spatially distant genomic elements. For each input length, we trained several models, increasing the kernel size by 10 from 5 to the size of that model's input. We report the Pearson correlation for each such experiment (Fig 3A). We found that increasing the size of the kernel resulted in improved prediction performance, and that the results reached a plateau as the kernel size approached the size of the model input. While increasing the number of parameters carries with it the risk of overfitting, as the number of kernels is only 256, it is unlikely that the model memorized the training data. When selecting the optimal model, we have taken into account the trade-off between performance and training time, leading us to select 100 nt-long flanks with a kernel size of 95.

Last, we tested which of the flanks of the 15 nt-long central bin was more important for prediction performance. For the selected flank length and kernel size, we trained two additional models. In each one we eliminated one of the flanks, i.e. 5' or 3', leaving the input with a flank either upstream or downstream the 15 nt-long bin. We observed that for the K$^+$ dataset, the achieved Pearson correlation on the validation set of chromosome 2 dropped from 0.82 to 0.68 and 0.4, with the upstream or downstream flanks only, respectively. For the K$^+$+PDS dataset, the achieved Pearson correlation dropped from 0.91 to 0.79 and 0.49, respectively. We conclude that there is a very strong asymmetry in the mismatch mechanism, where most of the mismatch-associated information is upstream of the 15 nt-long bin rather then downstream. This is more likely a result of a technological artifact of the G4-seq protocol rather than a principle of the G4-folding mechanism as G4s have a helical shape and as we tested the flanks of the 15 nt-long bins rather than the flanks of the G4 structures.

**G4mismatch prediction performance evaluation on a held-out test set.** With the selected flank length of 100 nt and kernel size 95, we tested G4mismatch on the held-out test set, which is chromosome 1. G4mismatch predictions were highly correlated with the experimental measurements (Fig 3B). A Pearson correlation of 0.84 was achieved for the K$^+$ stabilizer and 0.91 for K$^+$+PDS (S1(B) Fig). While several methods were already developed based on the G4-seq dataset [28, 30, 32, 33, 47], no other method has been used for a genome-wide mismatch-score prediction. Thus, as the only method for the task we set the baseline

accuracy of 0.84 and 0.91 Pearson correlation for the $K^+$ and $K^+$+PDS datasets, respectively. In addition, we evaluated the prediction performance of G4mismatch in a balanced setting. We binned chromosome 1 samples to three uniform bins by their mismatch scores and randomly sampled 1,000 samples from each bin. We repeated this sampling procedure 1,000 times and calculated the achieved correlation each time. G4mismatch achieved a high average correlation in the balanced setting: $0.892 \pm 0.005$ and $0.923 \pm 0.003$ for $K^+$ and $K^+$+PDS models, respectively (S2 Fig).

To evaluate the performance of G4mismatch over G4-forming sequences, we filtered sequences that fall under the extended definition of PQs [30]: $G^{3+}[ACGT]^{1-12}G^{3+}[ACGT]^{1-12}G^{3+}[ACGT]^{1-12}G^{3+}$ in chromosome 1, excluding those which contained PQs within the 100-nt downstream or upstream flanks, and PQs which lack experimental mismatch scores by G4-seq, and predicted their mismatch score for both $K^+$ and $K^+$+PDS. G4mismatch achieved a Pearson correlation of 0.87 and 0.81 on the $K^+$ and $K^+$+PDS PQ-only datasets, respectively, outperforming DeepG4, PENGUINN, Quadron, G4Hunter, pqsfinder, and g4predict (S3(A) and S3(B) Fig).

**G4mismatch prediction performance evaluation on an independent microarray dataset.** Validating the performance of G4mismatch on data from an experimental methodology other than G4-seq may prove that G4mismatch learned actual G4-related properties rather than biases or artifacts of the G4-seq protocol. We used G4mismatch to predict G4 propensity of previously microarray-measured G4 and non-G4 probe sequences. To do so, we padded each microarray probe sequence of original length of 60 nt by 100 nt on each side to form a sequence of 260 nt, and broke it into all possible 215 nt-long subsequences (the input size for G4mismatch). We predicted a mismatch score for each one. The overall sequence was assigned the maximum mismatch score predicted for its subsequences.

G4mismatch predictions were highly correlated with the PDS-binding intensities measured in the microarray experiment. A Pearson correlation of 0.81 and 0.78 was achieved by the models trained on $K^+$ (Fig 4A) and $K^+$+PDS (Fig 4B) stabilizers, respectively. The stabilizing effect PDS has on G4 structures led to overall higher mismatch scores predicted by the model trained on the $K^+$+PDS G4-seq dataset compared to the model trained on the $K^+$ dataset. Moreover, G4mismatch $K^+$+PDS model predictions have lower variability in the top bound

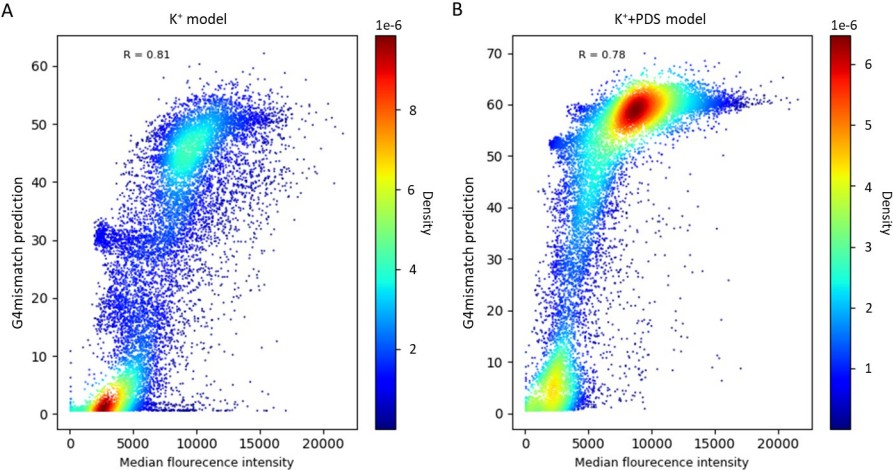

**Fig 4. G4mismatch predictions correlation to PDS-binding microarray data.** G4mismatch predictions highly correlate with with measurements of PDS-binding produced by a microarray experimental with both (A) $K^+$ and (B) $K^+$+PDS models.

microarray sequences since the model predicted high scores (>50% mismatch score) for all of them.

## Whole-genome G4 detection evaluation

We next aimed to utilize G4mismatch predictions to identify G4-forming sequences genome-wide. This would enable us to computationally identify G4s of varying lengths in a post-processing step to predicting mismatch scores to 15 nt-long bins. To test the G4 detection ability, we used G4mismatch to predict mismatch scores to all 15 nt-long bins of the held-out test set that is chromosome 1. From the predicted scores of chromosome 1 we identified G4 hit regions. We then assigned a binary label to each one of the nucleotides, 1 for a nucleotide belonging to a sequence identified as a G4 and 0 otherwise. G4s identified by the G4-seq method in chromosome 1 were used as ground truth and labeled similarly. We then used precision, recall, and Jaccard metrics to evaluate G4 detection performance. The achieved scores were 0.73, 0.71, and 0.56, for each of the metrics, respectively, over the $K^+$ dataset (Fig 5) and 0.83, 0.76, and 0.66 over the $K^+$+PDS dataset (S4 Fig).

**Comparison of G4 detection performance to extant methods.** We compared G4mismatch in G4 detection to six state-of-the-art methods: DeepG4, PENGUINN, Quadron, G4Hunter, pqsfinder, and g4predict. G4mismatch achieved the best precision-recall balance and greatest similarity (as measured by Jaccard index) to the test set for both $K^+$ and $K^+$+PDS models (Fig 5 and S4 Fig, respectively). Quadron, which was originally trained using the $K^+$-stabilized PQs, presented slightly higher precision of 0.74 compared to G4mismatch. This might stem from the fact that Quadron's training was based only on canonical G4s, making its positive predictions highly accurate. Similarly to Quadron, g4predict, which was designed for PQ prediction, achieved relatively high precision of 0.53. But, both Quadron and g4predict suffer from low recall, scoring 0.15 and 0.1, respectively, and coming in last out of all the methods. PENGUINN(s), on the other hand, showed higher recall than the other extant methods, scoring 0.79, and outperformed G4mismatch on the $K^+$ dataset, but this came at the expense of a low precision score of 0.01. Increasing the cut-off threshold by PENGUINN(p) did not improve any of the metrics compared to PENGUINN(s), and resulted in a substantial decrease in recall to 0.34. G4Hunter and pqsfinder also suffer from an imbalance between recall and precision, where G4Hunter had better recall of 0.53 while pqsfider had better precision of 0.24, but overall were outperformed by G4mismatch in all three metrics. DeepG4, as a cell-type specific method, failed to achieve accurate predictions, achieving a Jaccard similarity score of only 0.03.

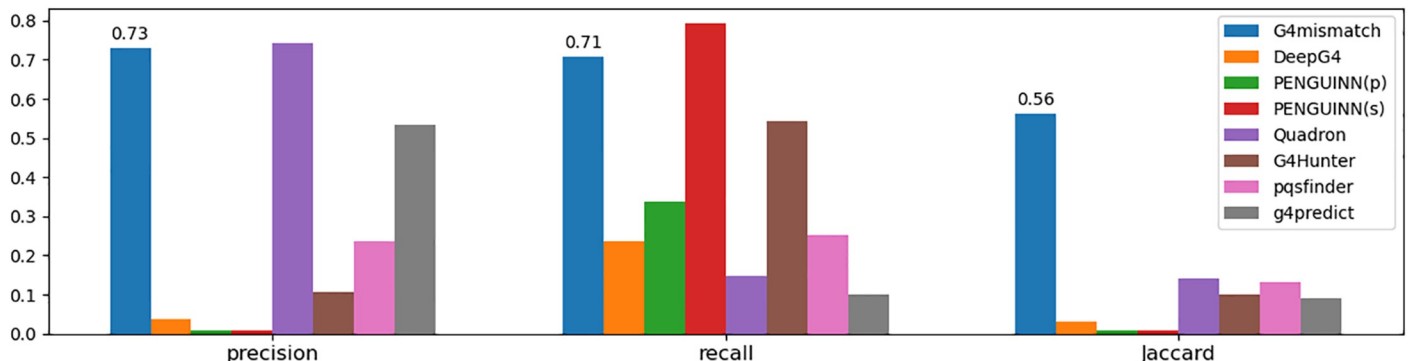

**Fig 5. G4mismatch and extant methods performance in $K^+$-stabilized G4 detection.** G4mismatch has consistent superior performance as evaluated by precision, recall, and Jaccard scores, achieving high recall without compromising precision.

### Inter-species prediction performance evaluation

We took advantage of the recently published G4-seq datasets of 12 species to test if G4-folding principles learned from one species can transform to other species. To answer the question, we trained G4mismatch on each of the species and tested each species-specific model on all other species-specific datasets.

Results show that some species-specific models were much more accurate than other models (Fig 6A and S5(A) Fig), and some species datasets are easier to predict than others. The human, mouse, and zebrafish models exhibited the best average performance, achieving an average Pearson correlation of 0.61, 0.61, and 0.60, respectively, on the 11 other species for each model trained on the $K^+$ dataset. Surprisingly, the Leishmania-specific model achieved excellent performance when predicting mismatch scores in other species, even, in some cases, outperforming species, which are phylogenetically more related. For example, the Leishmania-specific model achieved a Pearson correlation of 0.72 on the human $K^+$ dataset, compared to 0.66 achieved by *C. elegans* model. In contrast, some species exhibited poor performance. For example, the bacteria Rhodobacter model achieved a mean Pearson correlation of 0.25. The worst performance was obtained by another bacterial model, the *E. coli*-specific model, which achieved very low correlations of around -0.02. A recent study revealed a unique and non-random localization of G4-forming sequences in bacterial genomes, which may partially explain the performance of the bacterial-specific models [48].

We next asked what are the properties that correlate with mismatch-score prediction performance. To answer this question, we calculated the Spearman correlation of each of model performance vectors to six properties of the training datasets: genome size, GC-content, number of PQs, PQ density, frequency of mismatch scores under 5%, and single-nucleotide coverage (Fig 6B and S5(B) Fig). The properties with a significant correlation with model performance were genome size, number of PQs, and PQ density (harmonic mean p-value<0.007 and <0.002 for

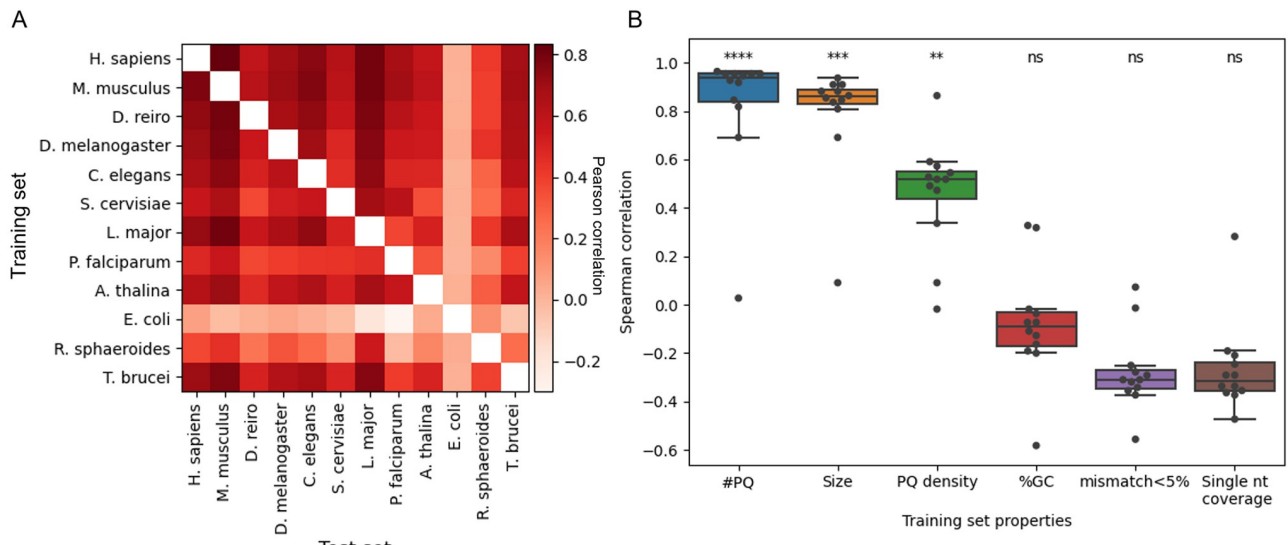

**Fig 6. Inter-species G4mismatch analysis and correlation to genomic and experimental properties.** (A) Inter-species G4mismatch $K^+$ prediction performance. Each model was trained on data of one species, and tested on all other species. Prediction performance is reported in Pearson correlation of predicted and measured G4 mismatch scores. (B) Correlations between G4mismatch models and number of PQs (#PQs), genome size, PQ density, GC-content (%GC), frequency of mismatch scores under 5% (mismatch<5%), and single-nucleotide coverage of the training datasets. For each property, prediction results were Spearman-correlated to model performance in Pearson correlation. P-value notation: ns = non-significant, ** ≤ 0.009, *** ≤ 0.001, **** ≤ 0.0001, adjusted for harmonic mean p-value.

K[+] and K[+]+PDS, respectively). The non-significant correlated properties were GC-content, frequency of mismatch scores under 5%, and single-nucleotide coverage. We speculate that properties related to the experimental data quality may explain part of the variability in prediction performance. However, since we could only extract one such property, and it is only partially correlated to data quality, i.e. single-nucleotide coverage, which is an indicator of the sequencing depth and quality, we found it to be of lesser significance to model performance compared to inherent properties of the genomes. In addition, we believe that the inherent genomic properties may had a more significant effect since machine-learning models benefit from having many data points of medium quality more than having few data points of high quality. The GC-content also has lesser significance for model performance, as it is probably the guanine content which is an important property for mismatch-score prediction, as can be deduced from the high correlation with the PQ-related properties. This might explain the poor performance of the model trained on *E. coli* data, which on the one hand, has relatively high GC content (50.8%) and single-nucleotide coverage (third highest on both parameters out of the 12 species), but on the other hand, has the lowest PQ count of all species measured in a G4-seq experiment. The frequency of mismatch scores under 5% is negatively correlated with models' performance, in contrast to PQ related features (number of PQs and PQ density), which are associated with higher mismatch scores, and have stronger positive correlation with models' performance.

## G4-folding principles learned by G4mismatch

We inspected the effect of loop lengths, mismatches in the G-tracts, and nucleotide composition of flanking sequences on predicted mismatch scores. Varying the length of the loops of the sequence of shape GGGNGGGNGGGNGGG from 1 nt to 12 nt showed a different behavior in the K[+] model compared to K[+]+PDS model (Fig 7A). G4mismatch K[+] model predicted a decrease in mismatch scores with the increase in loop lengths, especially with the inner loop compared to the two outer loops. This inverse association between loop length and G4 stability, which was learned by G4mismatch from G4-seq data, was previously observed in a low-throughput assay [49]. In contrast, the mismatch score predictions of G4mismatch K[+]+PDS model were barely affected by the increase in loop lengths. A minor decrease in mismatch scores was predicted after increasing the length of the inner loop from 6 nts to 8 nts. This stabilizing effect of PDS on G4 formation, independent of loop length, which was learned by G4mismatch from G4-seq data, was previously observed in microarray experiments [35].

Similar effects were observed when mutating G-tracts of GGGNGGGNGGGNGGG for both K[+] and K[+]+PDS models (Fig 7B and S6(A) Fig, respectively), which resulted in a decrease in the predicted mismatch score. Mutations in the second G-tract induced the greatest decrease compared to the canonical G4, which is, as far as we know, a novel finding, which should be further experimentally validated. We expected to see a higher effect of mutations in the middle G-quartet on G4 destabilization, which was previously observed in a low-throughput experiment [50]. We speculate that this was not observed in our visualization of G4mismatch-learnt G4-folding principles as a consequence of a technological artifact of the G4-seq assay, which should be further studied for thorough mitigation.

We measured the effect of nucleotide composition within the flanking sequences of a 15 nt-long of the form GGGNGGGNGGGNGGG on mismatch-score prediction (Fig 7C and S6(B) Fig). G4mismatch presented higher sensitivity to variations in the upstream flanking sequence, compared to the downstream flanking sequence. Increasing the probability of cytosine impaired the ability of the sequence to form a G4 much more than increasing the probability of guanine improved the ability of the sequence to form a G4. For example, setting the probability of cytosine to 1, while the other bases had a probability of 0, in the range of 20 nt

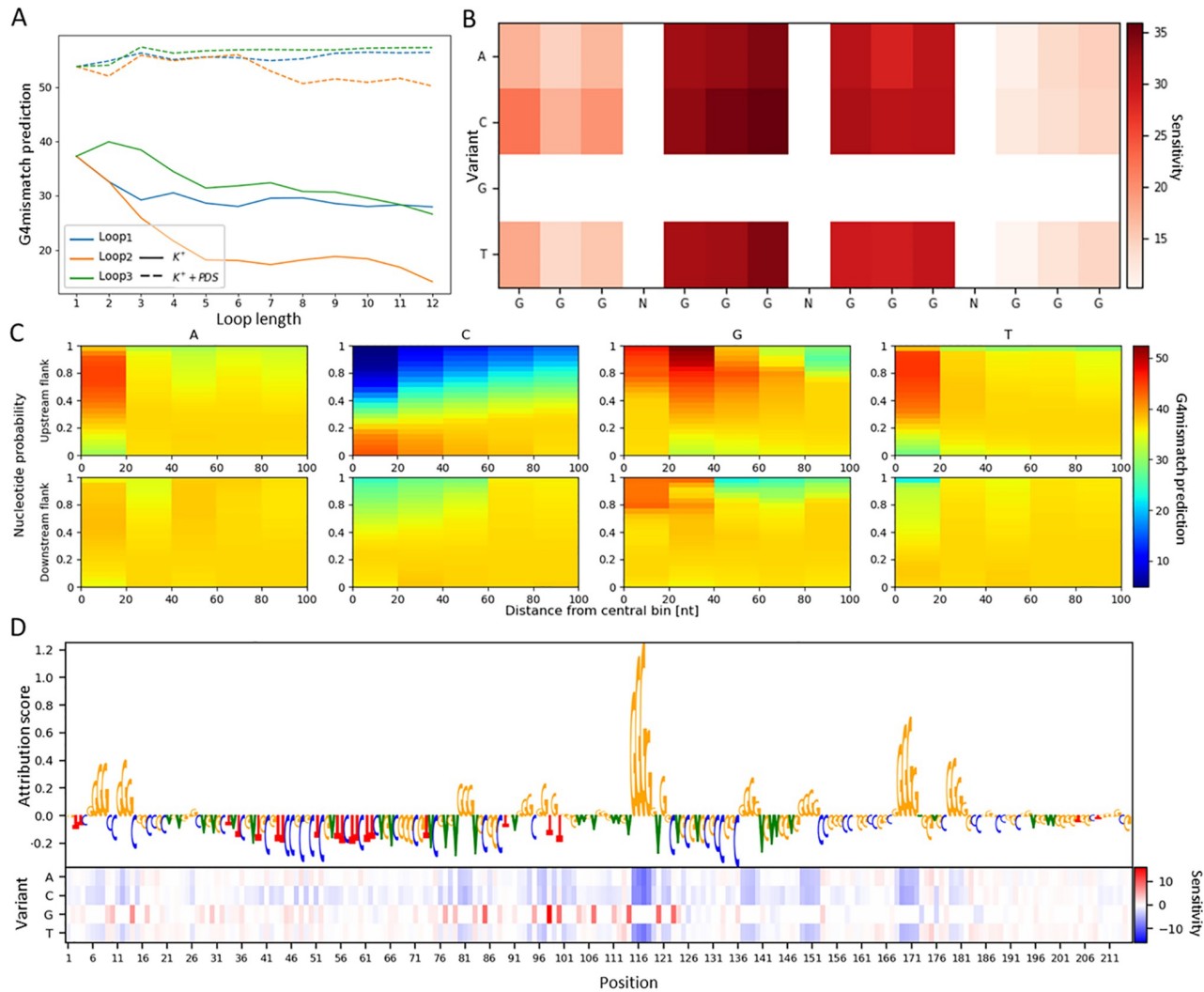

**Fig 7. Interpreting G4mismatch.** (A) Loop length effect on mismatch-score prediction. For the canonical G4 GGGNGGGNGGGNGGG, we varied the length of each loop separately, and predicted the mismatch score using both the $K^+$ and $K^+$+PDS models. (B) Effect of mismatch in the G-tracts on $K^+$ model predictions. For the canonical G4 GGGNGGGNGGGNGGG, we mutated each nucleotide in the G-tracts separately to all other nucleotides, and calculated the change in the predicted mismatch score. (C) Effect of nucleotide composition in the flanking sequences on $K^+$ model predictions. For the canonical G4 of the form GGGNGGGNGGGNGGG, we varied the probability of each nucleotide at a time, while assigning uniform probabilities to the other three nucleotides, in 20nt-long regions away from the central bin. We predicted the mismatch score for each such variant, on both the upstream and downstream flanks, separately. (d) The mutation map shows the sensitivity of the $K^+$ model to mutations and the corresponding attributions report the importance of a given feature to the model's prediction.

upstream from the central 15 nt-long bin resulted in a decrease of 32.44 in the predicted mismatch score, compared to the prediction for sequences consisting of equal probability for all nucleotides, whereas a probability of 1 of guanine within that same region increased the predicted mismatch score by only 9.47. We speculate that at a certain point the stability of a potential G4 achieves certain saturation, and that adding more guanines to an already stable structure, i.e. a canonical G4, has a little effect on its stability, while adding cytosines increases the probability of the guanines to base-pair. Adenine and thymine behaved similarly to each other, and mostly had a rather neutral effect on G4mismatch's predictions. Modifying the probability of adenine or thymine in the region closest to the central 15 nt-long bin increased

the predicted mismatch score; this is probably due to the fact that increasing their probability corresponds to a decrease in cytosine probability.

Last, we used the IG method to visualize the key features identified by G4mismatch. Fig 7D and S6(C) Fig portray the attribution scores calculated for the maximum scoring 15 nt-long sub-sequence and its flanks of the KRAS proto-oncogene using $K^+$ and $K^+$+PDS models, respectively. The attribution scores indicate that G4mismatch prediction is based on G-rich sub-sequences. The relevant guanines reside in continuous stretches of tall letters. The corresponding mutation map supports the results achieved by the IG method, showing that the model is highly sensitive to mutation of a given nucleotide from or to guanine. The highest attribution scores are assigned to G-stretches in the adjacent positions downstream of the central bin (positions 115–118), despite the fact that overall the downstream flank is less informative, as we deduced in the flank analysis (Fig 7C). This can be explained by the fact that IG and mutation map approaches are interpreting the G4-folding rules of only a single G4 example.

## Discussion

Computational prediction of G4s is a long-standing challenge in G4 research. Various computational methods were developed throughout the years, but only recently machine-learning models were trained on the high-throughput data collected by new experimental techniques [21]. Still, no method to date was developed to predict mismatch scores in a genome-wide manner, and use them to explore the potential of forming a G4 structure for any given genomic sequence. The developers of Quadron stated that "Having the complete G4-seq profile for the human genome, we were first tempted to develop a general machine learning model to target the link between any DNA sequence and G4-seq measured mm%. However, the approach did not result in a model with significant predictive power in validation tests." The goal of G4mismatch is to computationally simulate the exact conditions that led to the results generated by the G4-seq experiment, which is considered as the state-of-the-art experimental assay to measure G4s. To overcome the noisy mismatch scores, the same approach used to calibrate the scores from a given G4-seq experiment can be applied to the predictions made by G4mismatch.

In G4mismatch we made a breakthrough in the ability to recapitulate a G4-seq experiment and provide a mismatch score for any given DNA sequence with high accuracy. As we tested in a leave-chromosome-out setting over all 15 nt-long genomic bins, the Pearson correlation of our predictions achieved a score of 0.84 and 0.91 for $K^+$ and $K^+$+PDS, respectively. Furthermore, G4mismatch outperformed existing methods in both prediction of mismatch scores of PQs, and in a downstream G4 detection task. While Quadron and g4predict achieved high precision with low recall, and PENGUINN(s) achieved high recall with almost-zero precision, G4mismatch achieved both high precision and recall, leading to a high Jaccard score. This demonstrates the superiority of G4mismatch in both sensitivity and specificity in the G4 detection task. Future work may include a more extensive search of the hyper-parameter space of the G4mismatch model, and the utilization of more advanced deep neural network architectures, such as transformers and attention layers [51].

The validation on independent G4-seq datasets not only demonstrated the accuracy of G4mismatch predictions, but also strengthened the validity of G4-seq measurements. As G4-seq measurements were produced on different species, a systematic comparison of the G4 principles identified in one species compared to another was not possible. Our approach of learning a model to generalize over measurements produced on one species and using the model to predict G4 formation over a genome of a different species is the first systematic comparison. The high Pearson correlation values, between 0.42 and 0.83 and between 0.74 and

0.89 in most inter-species correlations achieved by the human $K^+$ and $K^+$+PDS trained models, respectively, strengthen the validity of G4-seq measurements.

A limitation of G4mismatch lies in its 'black-box' nature. This limitation is common to all DNN-based methods. While DNN-based methods are very successful in approximating complex functions, and thus achieve high accuracy on different predictions tasks, they are not interpretable as classic machine-learning methods, where features are manually-engineered and easily understood. We provide a partial remedy to this limitation by generating attribution-score plots of a G4 test sequence and observing the important features. Moreover, to gain further knowledge of G4mismatch-learnt principles of the G4-folding mechanism, we explored the predictive outcomes of several variations of a canonical G4. G4mismatch was able to uncover known principles as well as novel ones. The uniqueness in our visualization approaches stems from our choice to encode our inputs as local nucleotide distributions. In this encoding, we varied the weight of the four nucleotides in each position, such that the sum of the weights is one, allowing it to represent a distribution over each position, rather then one-hot encoding of specific sequences. In this way, we efficiently represent a variety of sequences. The fact that the insights we derived from the G4mismatch-trained model are in concordance with prior knowledge further strengthens the validity of our unique approaches. Our investigation of G4-folding principles is certainly not comprehensive enough to decipher G4mismatch model complexity, but the problem of interpreting DNNs is a general open problem in the field, not specific to our study. Further validation and exploration of the G4mismatch model may be conducted on experimental data, such as the Genotype-Tissue Expression (GETx) project [52], to derive the functional effects of mutations in G4s on gene expression.

## Conclusion

In this work, we presented G4mismatch, the first method to predict a mismatch score to any given genomic DNA sequence based on G4-seq data. In addition to taking advantage of the recently high-throughput produced datasets, we utilized the most advanced computational learning techniques, i.e. DNNs. We expect G4mismatch to be in broad use in G4 research to identify sequences with the potential to form G4 structures.

## Supporting information

**S1 Table. Number of G4 mismatch scores reported in each of the G4-seq datasets used this study.**
(PDF)

**S1 Fig. G4mismatch $K^+$+PDS model prediction performance.** (A) For each flanking sequence length, multiple models were trained differing in their kernel size. Increasing the kernel size resulted in improved performance, which plateaued at a kernel size of the flank length. (B) G4mismatch predictions on chromosome 1 test set highly correlate with G4-seq experimental measurements.
(TIF)

**S2 Fig. G4mismatch prediction performance on balanced test sets.** We uniformly binned the held-out test set to 3 bins, and randomly sampled 1,000 samples from each bin over 1,000 iterations to generate balanced test sets. Pearson correlations between predicted and measured mismatch scores over the balanced test sets are reported.
(TIF)

**S3 Fig. G4mismatch and extant methods performance in prediction of mismatch scores of PQs.** Results on held-out chromosome 1 for (A) $K^+$ and (B) $K^+$+PDS stabilizers. We extracted all sequences under the definition of an extended PQ, i.e. $\{G^{3+}[ACGT]^{1-12}\}^{3+}G^{3+}$, excluding those which contained PQs within the 100-nt downstream or upstream flanks, and PQs which lack experimental mismatch scores by G4-seq.
(TIF)

**S4 Fig. G4mismatch and extant methods performance in $K^+$+PDS-stabilized G4 detection.** G4mismatch has consistent superior performance as evaluated by precision, recall, and Jaccard scores, achieving high recall without compromising precision.
(TIF)

**S5 Fig. Inter-species G4mismatch analysis and correlation to genomic and experimental properties.** (A) Inter-species G4mismatch $K^+$+PDS prediction performance. Each model was trained on data of one species, and tested on all other species. Prediction performance is reported in Pearson correlation of predicted and measured G4 mismatch scores. (B) Correlations between G4mismatch models and number of PQs (#PQs), genome size, PQ density, GC-content (%GC), frequency of mismatch scores under 5%, and single-nucleotide coverage of the datasets each model was trained on. For each property, prediction results were Spearman-correlated to model performance in Pearson correlation. P-value notation: ns = non-significant, ** $\leq 0.009$, *** $\leq 0.001$, **** $\leq 0.0001$, adjusted for harmonic mean p-value.
(TIF)

**S6 Fig. Interpreting G4mismatch.** (A) Effect of mismatch in the G-tracts on $K^+$+PDS model predictions. For the canonical G4 GGGNGGGNGGGNGGG, we mutated each nucleotide in the G-tracts separately to all other nucleotides, and calculated the change in the predicted mismatch score. (B) Effect of nucleotide composition in the flanking sequences on $K^+$+PDS model predictions. For the canonical G4 of the form GGGNGGGNGGGNGGG, we varied the probability of each nucleotide at a time, while assigning uniform probabilities to the other three nucleotides, in 20nt-long regions away from the central bin. We predicted the mismatch score for each such variant, on both the upstream and downstream flanks, separately. (C) The mutation map shows the sensitivity of the $K^+$+PDS model to mutations and the corresponding attributions report the importance of a given feature to the model's prediction.
(TIF)

## Acknowledgments

We gratefully acknowledge the support of NVIDIA Corporation with the donation of the TITAN V GPU used for this research.

## Author Contributions

**Conceptualization:** Yaron Orenstein.

**Funding acquisition:** Yaron Orenstein.

**Investigation:** Mira Barshai, Barak Engel, Idan Haim.

**Methodology:** Mira Barshai.

**Software:** Mira Barshai.

**Supervision:** Yaron Orenstein.

**Validation:** Mira Barshai.

**Visualization:** Mira Barshai.

**Writing – original draft:** Mira Barshai, Yaron Orenstein.

**Writing – review & editing:** Mira Barshai, Yaron Orenstein.

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
