## [Decision Letter · Decision Letter 0]

6 Feb 2022

Dear Dr Orenstein,

Thank you very much for submitting your manuscript "G4mismatch: deep neural networks to predict G-quadruplex propensity based on G4-seq data" for consideration at PLOS Computational Biology.

As with all papers reviewed by the journal, your manuscript was reviewed by members of the editorial board and by several independent reviewers. In light of the reviews (below this email), we would like to invite the resubmission of a significantly-revised version that takes into account the reviewers' comments.

We cannot make any decision about publication until we have seen the revised manuscript and your response to the reviewers' comments. Your revised manuscript is also likely to be sent to reviewers for further evaluation.

Sincerely,

Faisal Mohamed Fadlelmola, Ph.D.

Guest Editor

PLOS Computational Biology

Ilya Ioshikhes

Deputy Editor

PLOS Computational Biology

Reviewer's Responses to Questions

**Comments to the Authors:**

Reviewer #1: I would like to thank the editor for inviting me to review this research article, titled G4mismatch: Deep Neural Networks to Predict G-Quadruplex Propensity Based on G4-seq Data

The article is well written. The authors have optimised the G4mismatch tool for the best performance by fine-tuning different parameters. It uses existing G4s data for 12 species to train deep neuronal networks and evaluate them against available other data from another research project and cross-species comparisons (trained with one species and tested with another species). The article also reports on the effects of the length of loops, nucleotide composition in flanking regions and mismatches in G4 structures on the predictability of models under two different conditions K+ and K++ PDS. Authors report that G4mismatch performs better compare exiting tools.

Minor comments

1. Figure 7a: In Annotation both K+ and K++PDS are presented by `--` (dashed lines). I guess, dashed line belong to K++PDS.

2. It would interesting see the comparison with other new tools like g4predict (Pubmed id: 34650044) and DeepG4 (Pubmed Id: 34383754).

3. Please include example datasets in the G4mismatch git-hub repository.

Reviewer #2: In this manuscript, the Authors propose a CNN-based approach to predict mismatch scores in a genome-wide manner, which would be useful in detecting G-quadruplex structure formation. The Authors used data coming from G4-sequencing experiments, where the mismatch scores were measured for each mapped 15 nucleotide bin throughout the genomes of multiple species.

Major comments

1) In numerous places in the work, the Authors bring up “… no other method was developed for genome-wide prediction of mismatch scores”. Note, that a number of (at least three) methods developed before, utilized genome-wide G4-seq data, but surely none that would then predict mismatch scores throughout the whole genome, rather than at only G-rich sites.

The contrast that this work brings by focusing on all the genomic locations to predict mismatch scores, is however dangerous, and should be thoroughly reconsidered or validated by means other than brought in the work.

The human genome contains ~200 times more 15nt bins that do not form G4s, as compared to the bins that take part in G4 formation. The underlying experimental methodology (G4-seq) is based on erratic, unclea, base calling behavior that happens when the polymerase stalls. The resulting errors, that manifest as mismatch scores, are very much dependent on how a sequencer treats uncertainties upon stalling, and how the base callers behave in such conditions. This behavior could be arbitrarily dependent on the exact sequencer model, base caller software version, and conditions, and is not something that one would normally wish to predict. G4-seq experiments, however, then go on to calibrate the methodology from a given experiment, to find the G4 structures specifically, differentiating those signals from the underlying “noise”. The proposed G4mismatch model nearly “aims” to predict that underlying noise (with ~200 times more data for that background noise, as compared to actual G4 signals). I am not sure how useful this model in general would be, and how consistent it would be if the sequencer model in G4-seq experiments change.

Moreover, the 200-times-more noise part of the genome will mostly have near-0 mismatch scores, as compared to around 40 mismatch scores for actual G4 signals (see y-axis in Figure 4). This type of tight clustering of values at around 0 and 40 will result in artificially high correlation coefficients, hence I would be very cautious in evaluating any of the model performance and validation outcomes from such an unbalanced data representation. While comparing multiple species, the size of the genome might be an important determinant in the success of comparisons (see Figure 6b), by simply increasing the stretch of near 0 mismatch scores.

2) As the core validation methodology in the method development and hyperparameter optimization, the Authors used leave-a-chromosome out approach. This however is not correct taking into account a large amount of sequence-wise similar G4s and repeat elements shared between chromosomes. Therefore, there can be a very large amount of sequences in the left out chromosome 2, the core (most influential central 15nt and peri 15-nt) part of which would be very close to the sequences already present in the training set. The training set and the machine learning model, in this setup, could simply overfit the data, arriving to a wrong selection of hyperparameters presented in the work. Later, in the caption of Figure 3 we have “Increasing the kernel size resulted in improved performance, which plateaued at a kernel size of the flank size”. This could mean that no extra abstraction of the sequence was necessary in this setup, and just directly using the original sequence information was resulting into the best performance. Could this be because of the leave-a-chromosome-out validation?

3) Part of the work discusses the importance of the flanks, and the hierarchy of the 5’ and 3’ flanks in defining the mismatch scores. However, the results have little genomic and G-quadruplex relevance, considering that by flanks the Authors meant flanks near a 15nt stretch, rather than flanks near exact G4 structures. The chosen flank ranges and 5’/3’ hierarchies may therefore reflect more of their influence on “background-noise” mismatch scores, and the hierarchy interpretation would be impeded by understanding how much each of those flanks bring the remaining bits of G4 sequence, normally longer than 15nt.

Further comments

a) The work will benefit from language revision and correction of numerous typos. Examples are “mismatch the scores” (Abstract), should be “mismatch scores”, “quadruplexs” (Introduction), G4hunter, should be “G4Hunter” as elsewhere (Introduction), “gneome” and “memeory” (section 2.3) and so on.

b) Please, replace “as” in G4s are formed “in the gnome as Hoogsteen bonds between guanines…”, with “facilitated by Hoogsteen bonds” (Abstract and Introduction).

c) In the Author Summary, we can read:

“Previous methods to predict G4s did not take advantage of the genome-wide information provided by the G4-seq experiment, but rather simply solved the problem of G4-folding as binary classification. Our new approach, G4mismatch, is the first to utilize millions of G4 mismatch scores …”

This statement is not true, as a number of machine learning-based methods exist that are based on G4-seq, in part discussed in the same manuscript.

Reviewer #3: Thank you for the opportunity to review the manuscript: “G4mismatch: deep neural networks to predict G-quadruplex propensity based on G4-seq data". G-quadruplexes have been demonstrated to be important NA structures required for regulation of basic biological processes. Although there are many algorithms for their prediction, the characterization of active G-quadruplexes remains a challenge. Therefore, new approaches to address this issue are very desirable and welcome. The manuscript contains interesting data; however, crucial citations are mission into the Introduction. The Discussion is very short and unappropriated. The comparisons between species are very interesting; however, the raw data are not presented in the results, so it is impossible to verify authors statements.

Where are the raw data analysis? Where is the overlap of individual sequences with the experimental data? Provide the list of sequences identical to G4-seq, what are different - they are not in G4seq datasets and vice versa?

The G4mismatch score is not described accordingly. There is described only as a comparison to experimental data, but how is used to determine in uncharacterized sequence? What do the numbers on the y-axis in Figure 2b mean? What are the theoretical maximal and minimal scores of G4mismatch?

I cannot see the mention of 'unique visualization' (?) – Show and describe your 'unique visualization' with examples for various sequences. (chapter 2.7)

The discussion lacks analysis of results and comparison with other algorithms, what false positives and false negatives for G4mismatch and with comparison with other tested algorithms?

The crucial current literature according to algorithms for G-quadruplex predictions and about G-quadruplex analyses in various species is missing (e.g. A guide to computational methods for G-quadruplex prediction, Nucleic Acids Res., 2020 Jan 10;48(1):1-15., doi: 10.1093/nar/gkz1097., The regulation and functions of DNA and RNA G-quadruplexes, Nat. Rev. Mol. Cell Biol., vol. 21, no. 8, pp. 459–474, Aug. 2020, doi: 10.1038/s41580-020-0236-x., The Presence and Localization of G-Quadruplex Forming Sequences in the Domain of Bacteria, Molecules, vol. 24, no. 9, May 2019, doi: 10.3390/molecules24091711, Landscape of G-quadruplex DNA structural regions in breast cancer, Nat. Genet., vol. 52, no. 9, pp. 878–883, Sep. 2020, doi: 10.1038/s41588-020-0672-8, etc.)

Please correct typos, for example: Page 4 – second line: „the reference gneome were stored in memeory“

**Have the authors made all data and (if applicable) computational code underlying the findings in their manuscript fully available?**

Reviewer #1: Yes

Reviewer #2: Yes

Reviewer #3: **No: **raw data are not shown

PLOS authors have the option to publish the peer review history of their article (what does this mean?). If published, this will include your full peer review and any attached files.

Reviewer #1: No

Reviewer #2: No

Reviewer #3: No
---

## [Decision Letter · Decision Letter 1]

24 Aug 2022

Dear Miss Barshai,

Thank you very much for re-submitting your manuscript "G4mismatch: deep neural networks to predict G-quadruplex propensity based on G4-seq data" (PCOMPBIOL-D-21-01794R1) for consideration at PLOS Computational Biology. As with all papers peer reviewed by the journal, your manuscript was reviewed by members of the editorial board and by several independent peer reviewers. Based on the reports, we regret to inform you that we will not be pursuing this manuscript for publication at PLOS Computational Biology.

As follows from the comments of the new Reviewer #4, the authors didn’t address previous comments and recommendations in a proper way. 

The discussion section of the manuscript is unconvincing and rather brief. Although the species comparisons are quite intriguing, the results do not include the raw data, making it impossible to validate the authors' claims.

All these were issues criticized within the first round of reviewing.

The reviews are attached below this email, and we hope you will find them helpful if you decide to revise the manuscript for submission elsewhere. We are sorry that we cannot be more positive on this occasion. We very much appreciate your wish to present your work in one of PLOS's Open Access publications. 

Thank you for your support, and we hope that you will consider PLOS Computational Biology for other submissions in the future.

Sincerely,

Faisal Mohamed Fadlelmola, Ph.D.

Guest Editor

PLOS Computational Biology

Ilya Ioshikhes

Section Editor

PLOS Computational Biology

Reviewer's Responses to Questions

**Comments to the Authors: **

Reviewer #1: Authors have improved the manuscript “G4mismatch: Deep Neural Networks to Predict G-Quadruplex Propensity Based on G4-seq Data”based on reviewer comments. However, I would like to point out some minor issues.

1. On line 74, “ to generate whole-genome G4 maps for 12 species” can be removed.

2. Figure 2 legend, please chage “4mismatch” to “G4mismatch”.

3. Figure S1: Please change PDS to K+ + PDS

Reviewer #4: Summary:

The authors propose a novel deep learning algorithm to predict the percent of mismatch in G4seq experiment due to the presence of G4 formation on DNA. The article is well written and well supported by experimental results. Comparison with other algorithms shows the superiority of their approach. However, as far as I understood, the main contribution of their algorithm compared to previously published algorithms G4detector and PENGUINN, is that G4mismatch predicts a quantitative signal (the percent of mismatch), whereas the other algorithms predict mismatch peaks (binary classification). 

Major revisions:

- the authors wrote: "The G4 mismatch score, as defined by the developers of the

G4-seq assay, is calculated as the ratio of the number of mismatched base calls observed

under a G4-stabilizing condition compared to control conditions over the length of the

complete sequence (either mismatched or not). Hence, its range is 0% to 100%."

Later, they wrote: "The network’s output goes through a single neuron with linear activation.",

meaning that the model is aimed to predict a value ranging from minus infinity to plus infinity, while the mismatch score actually range from 0% to 100%. 

If I understood correctly, I believe a sigmoid activation as output would better suit, since the output is between 0 and 1. If no, the authors must explain this point. 

- In section "Inter-species prediction performance evaluation", the authors wrote that "Results show that some species-specific models were much more accurate than other models". Could it be partly due to the heterogenous quality of data between the different species? 

- the authors explored the impact of mutations on G4 based on their predictions. A simple way for the authors to experimentally validate their findings is to use human GTEx expression SNPs (eSNPs) and check what is the impact of their mutation on expression, given that a G4 in a gene promoter is known to regulate the gene expression. With my students, we have done already similar analyses, and there is some chance it will reveal some interesting results. 

- the authors propose an algorithm that is very similar to previously published algorithms G4detectors, PENGUINN and DeepG4, with a very classical CNN architecture. 

Given the recent progress in deep learning, the authors must try to use attention (Enformer) or deep residual CNN (Basenji) to improve their results. For instance, based on my experience, it is very likely that deep residual CNN with for instance 5 layers of CNNs with residual connections combined with dilated convolution with kernel size of 3 (or 5) will significantly improve the results (see Basenji).

Minor revisions:

- "The detected sequences are scored based the length of the tetrads" -> "The detected sequences are scored based ON the length of the tetrads"

- in the intro, you should explain that DeepG4 differs from other CNNs such as G4detector and PENGUINN, since it was trained (and aimed) to predict active G4s, meaning G4s formed in vitro (G4seq) which are active in vivo (G4 ChIP-seq) and thus cell type dependent. It thus explains why DeepG4 has lower accuracy when it comes to predict G4seq results, compared to G4detector and PENGUINN. It was actually explained in the results, line 369-370.

- in the intro, you should clearly state that G4mismatch predicts G4seq signal as a quantitative regression problem unlike G4detector and PENGUINN which predicts G4seq peak as a binary classification problem. 

- The authors wrote : "The input to G4mismatch is a one-hot encoded 215 nt-long DNA sequence matrix. N positions in the DNA sequence were replaced by a vector of 0.25 indicating a uniform probability to all nucleotides."

THen they wrote: "The uniqueness in our visualization approaches stems from our choice to

encode our inputs as local nucleotide distributions rather then one hot-encoding of

specific sequences."

I don't understand "local nucleotide distributions". Do the authors mean the weight attributed to each nucleotide when assessing its contribution to the G4 formation? Please explain.

**Have the authors made all data and (if applicable) computational code underlying the findings in their manuscript fully available?**

Reviewer #1: Yes

Reviewer #4: Yes

PLOS authors have the option to publish the peer review history of their article (what does this mean?). If published, this will include your full peer review and any attached files.

Reviewer #1: No

Reviewer #4: Yes: Raphael Mourad

---

## [Decision Letter · Decision Letter 2]

7 Feb 2023

Dear Miss Barshai,

We are pleased to inform you that your manuscript 'G4mismatch: deep neural networks to predict G-quadruplex propensity based on G4-seq data' has been provisionally accepted for publication in PLOS Computational Biology. Please address the minor correction raised by reviewer #1. 

Best regards,

Faisal Mohamed Fadlelmola, Ph.D.

Guest Editor

PLOS Computational Biology

Ilya Ioshikhes

Section Editor

PLOS Computational Biology

Reviewer's Responses to Questions

**Comments to the Authors:**

Reviewer #1: I would like to thank the authors for responding to my and other reviewer's comments.

Minor correction:

As many tools do not recognise as "" ext-link-type="uri" xlink:type="simple">github.com/OrensteinLab/G4mismatch/" (in astract) as web link, please change it to https://github.com/OrensteinLab/G4mismatch/.

In Tab1.pdf: please change Caenorhabditiselegans to Caenorhabditis elegans

Reviewer #4: The authors have addressed my comments.

**Have the authors made all data and (if applicable) computational code underlying the findings in their manuscript fully available?**

Reviewer #1: Yes

Reviewer #4: Yes

PLOS authors have the option to publish the peer review history of their article (what does this mean?). If published, this will include your full peer review and any attached files.

Reviewer #1: No

Reviewer #4: **Yes: **Raphaël Mourad

quillbot-extension-portal/quillbot-extension-portal

---

## [Editor Report · Acceptance letter]

3 Mar 2023

PCOMPBIOL-D-21-01794R2 

G4mismatch: deep neural networks to predict G-quadruplex propensity based on G4-seq data

Dear Dr Barshai,

I am pleased to inform you that your manuscript has been formally accepted for publication in PLOS Computational Biology. Your manuscript is now with our production department and you will be notified of the publication date in due course.

With kind regards,

Zsofia Freund
